# Targeted and tailored pharmacist-led intervention to improve adherence to antihypertensive drugs among patients with type 2 diabetes in Indonesia: study protocol of a cluster randomised controlled trial

Sofa D Alfian [1,2,3] Rizky Abdulah,[2,3] Petra Denig,[4,5] Job F M van Boven,[4,5] Eelko Hak[1,5]

For numbered affiliations see end of article.

**Correspondence to**
Sofa D Alfian;
sofa.alfian@unpad.ac.id

## ABSTRACT

**Introduction** Current intervention programme to improve drug adherence are either too complex or expensive for implementation and scale-up in low-middle-income countries. The aim of this study is to assess the process and effects of implementing a low-cost, targeted and tailored pharmacist intervention among patients with type 2 diabetes who are non-adherent to antihypertensive drugs in a real-world primary care Indonesian setting.

**Methods and analysis** A cluster randomised controlled trial with a 3-month follow-up will be conducted in 10 community health centres (CHCs) in Indonesia. Type 2 diabetes patients aged 18 years and older who reported non-adherence to antihypertensive drugs according to the Medication Adherence Report Scale (MARS) are eligible to participate. Patients in CHCs randomised to the intervention group will receive a tailored intervention based on their personal adherence barriers. Interventions may include reminders, habit-based strategies, family support, counselling to educate and motivate patients, and strategies to address other drug-related problems. Interventions will be provided at baseline and at a 1-month follow-up. Simple question-based flowcharts and an innovative adherence intervention wheel are provided to support the pharmacy staff. Patients in CHCs randomised to the control group will receive usual care based on the Indonesian guideline. The primary outcome is the between-group difference in medication adherence change from baseline to 3-month follow-up assessed by MARS. Secondary outcomes include changes in patients' blood pressure, their medication beliefs assessed by the Beliefs about Medicines Questionnaire (BMQ)-specific, as well as process characteristics of the intervention programme from a pharmacist and patient perspective.

**Ethics and dissemination** Ethical approval was obtained from the Ethical Committee of Universitas Padjadjaran, Indonesia (No. 859/UN6.KEP/EC/2019) and all patients will provide written informed consent prior to participation. The findings of the study will be disseminated through international conferences, one or more peer-reviewed journals and reports to key stakeholders.

### Strengths and limitations of this study

► The pharmacist-led intervention programme uses principles of targeting by screening for non-adherence and tailoring to the patients' personal adherence problems to enhance its potential effect. Simple question-based flowcharts and an innovative adherence intervention wheel are provided to support the pharmacy staff.

► The intervention aligns with the current workflow and resources in the daily clinical practice of a lower-middle-income country and will not require a substantial change to the current care system.

► This study is designed as a cluster randomised controlled trial to reduce the risk of bias and contamination across study groups.

► The implementation process will be evaluated using the RE-AIM framework, which will enable us to carefully interpret the results and guide us when scaling up the intervention to include a larger population in Indonesia and other lower-middle-income settings.

► As is common with many behavioural intervention studies, it is not possible to blind the researchers and pharmacists to the group allocation of patients.

**Trial registration number** NCT04023734.

## INTRODUCTION

In patients with type 2 diabetes, the pharmacological treatment of comorbid hypertension can substantially reduce the risk of cardiovascular complications.[1] However, although effective pharmacological treatment is available, adherence to antihypertensive medications in patients with type 2 diabetes is known to be suboptimal.[2] Notably, non-adherence to antihypertensive medications is associated with poor health

outcomes and increased healthcare costs.[3] Therefore, effective intervention strategies to enhance adherence are urgently required.

Previous studies showed that patients may not take their medication for various reasons. Non-adherence could arise following a conscious decision after balancing the pros and cons of medication (intentional non-adherence), could be due to a lack of understanding of the medication regimen or due to forgetfulness (unintentional non-adherence).[4–7] The reasons underlying intentional and unintentional non-adherence are not entirely independent and are heterogeneous. These reasons include lack of attention, lack of knowledge, high concerns and/or low necessity beliefs, which can reduce motivation.[4–7] In addition, there may be other drug-related problems, such as difficulties with intake or high costs, that can lead to non-adherence.[8 9] As such, there are no one-size-fits-all solutions to address non-adherence.

In developed countries with well-established healthcare systems, a wide variety of interventions to improve medication adherence have been developed.[10 11] Six main types of interventions can be identified: patient education, medication regimen management, clinical pharmacist consultation, cognitive-behavioural therapies, medication-taking reminders and incentives to promote adherence.[11] However, a Cochrane review showed that most interventions are often too complex and not particularly effective.[10] Additionally, in lower-middle-income countries, the paucity of healthcare and economic resources poses challenges to proper implementation of adherence enhancing interventions.

Non-adherence may be more efficiently improved if only patients who need it are targeted and interventions are tailored to patients' individual adherence barriers.[12 13] Previous studies further suggested that effective interventions to improve adherence were led by a pharmacist, delivered face-to-face, administered directly to patients and behaviourally targeted compared with cognitively targeted interventions.[14] In Southeast Asia, however, pharmacy services can be hampered by inadequate training of pharmacy staff.[15] Furthermore, adherence interventions may not be sustained over time due to a lack of resources to maintain them. Thus, to improve adherence to antihypertensive drugs in patients with type 2 diabetes, a low-cost, targeted and tailored pharmacist-led intervention that can be integrated into the community pharmacy workflow is required.

The primary objective of this study is to assess the effect of a targeted and tailored pharmacist-led intervention on medication adherence to antihypertensive drugs among patients with type 2 diabetes. The secondary objectives are (a) to assess the effect of the intervention on blood pressure level and medication beliefs, (b) to evaluate the implementation of the intervention from a pharmacist and patient perspective and (c) to assess the effects of the intervention across different subgroups of patients.

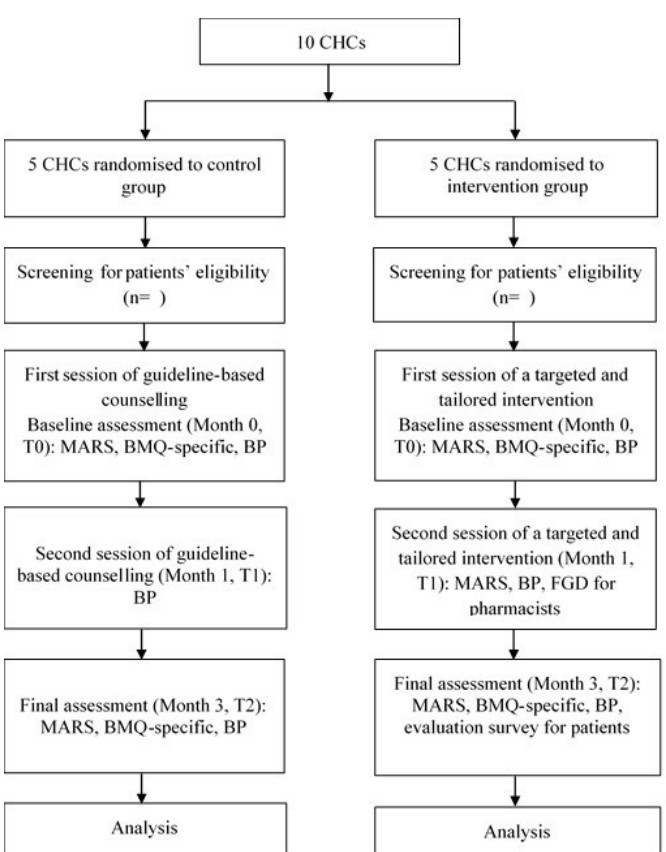

**Figure 1** Consort flowchart. BP, blood pressure; BMQ-specific, Beliefs about Medicines Questionnaire-specific; CHC, community health centre; FGD, focus group discussions; MARS, Medication Adherence Report Scale.

## METHODS AND ANALYSIS

This protocol was developed in accordance with the CONSORT 2010 statement for cluster randomised trials[16] and reported according to the SPIRIT checklist[17] (online supplementary appendix A). We will use the RE-AIM framework[18] to evaluate the implementation process of the intervention.

### Study design and setting

This study is a 3-month cluster randomised controlled trial with two parallel arms and will be performed in Bandung City, Indonesia from August to December 2019 (figure 1). Clusters of randomisation are community health centres (CHCs), locally called *Puskesmas*. *Puskesmas* are primary healthcare centres at the subdistrict level, with each centre staffed with medical doctors, nurses, midwives, and pharmacists.

### Participants

A total of 10 CHCs will be purposively selected based on a sufficient number of patients with type 2 diabetes with hypertension. The principal investigator (PI) will introduce and explain the study to the pharmacists and physicians in the CHCs. In each CHC, one pharmacist will be included. In Indonesia, one of the pharmacist's responsibilities during routine clinical practice is to counsel

patients with chronic diseases on their medication use, often performed in a counselling room that is separated from the drug counter.[19] Screening for patients' eligibility will be conducted by the pharmacist during regular outpatient visits. Once a patient is deemed eligible, the pharmacist will inform the PI or research assistant to approach the patient and briefly explain the study, and ask to sign informed consent (online supplementary appendix B). Patients will be eligible if they meet the following inclusion criteria: (i) at least 18 years old, (ii) diagnosed with type 2 diabetes for at least 1 year based on patient's medical record, (iii) using at least one antihypertensive drug in the last 3 months, (iv) have signed informed consent and (v) have suboptimal medication adherence to antihypertensive drugs according to the Medication Adherence Report Scale (MARS score <20; MARS scores range from 5 to 25). Of note, our previous work in four regions in Indonesia (Bandung City, Yogyakarta City, Makassar City and Samarinda City) showed that half of the patients were non-adherent, with a MARS score <20. Patients with severe mental or physical constraints, pregnancy or in the lactation period, illiterate in the Indonesian language, enrolment in another intervention study and those not responsible for taking their own medication will be excluded.

### Randomisation and blinding
We will use cluster randomisation at the CHCs level to reduce the risk of bias and contamination across study groups.[20] The PI will randomise the CHCs into the intervention or control group in a 1:1 ratio. The randomisation sequence will be generated using a random number generator. Given the nature of the study design, both pharmacists and the PI cannot be blinded to the group assignment.

### Intervention
Patients in the five CHCs randomised to the intervention group who were screened as non-adherent to their antihypertensive drugs will receive a tailored pharmacist-led intervention during two sessions (at baseline and at a 1-month follow-up) in addition to usual care. Both will be regular outpatient visits, when patients collect their medication. The intervention will be low cost, aligned with the current CHC workflow and will not require a substantial change to the current system. Before the study started, the intervention steps and materials were piloted in two pharmacies and optimised in an iterative process.

#### Intervention at baseline (first session)
Before dispensing antihypertensive drugs during the first session, the pharmacist will discuss patient-specific barrier(s) for medication adherence based on their responses to the MARS questionnaire and three additional questions, which are derived from the Brief Medication Questionnaire[21] (figure 2). The intervention strategy will then be tailored to the identified adherence problems. Based on the current literature,[22–24] we defined four non-adherence problems that can be addressed by the community pharmacists, that is, (1) forgetfulness, (2) lack of knowledge, (3) lack of motivation and/or (4) other drug-related problems. Of note, patients might need a combined intervention strategy to address all experienced problems. The four non-adherence problems

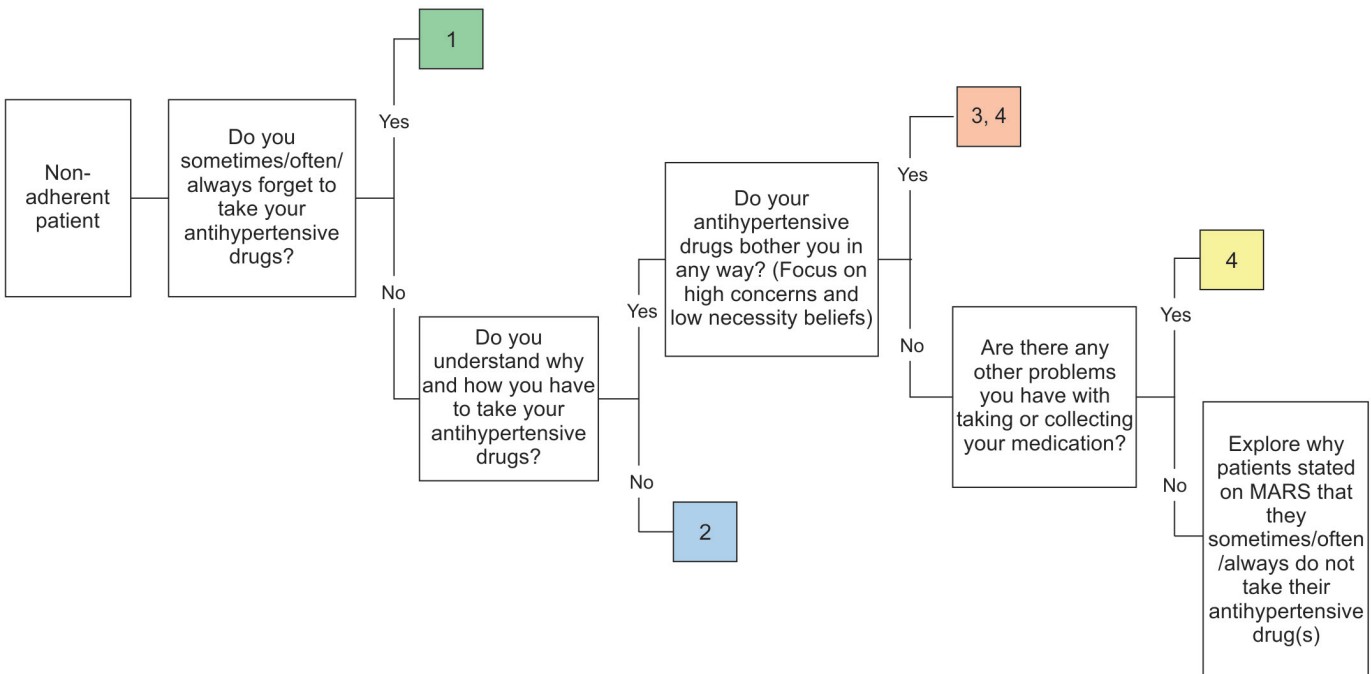

**Figure 2** Flowchart of a targeted and tailored intervention at baseline visit (T0). Type of intervention: 1=reminders, habit-based strategies and/or involvement of family member; 2=counselling to increase knowledge (teach-back method); 3=counselling to increase motivation; 4=explore/address other drug-related problems. MARS: Medication Adherence Report Scale.

Table 1 Non-adherence problems and recommended intervention strategies

| Non-adherence problems | Intervention strategies |
|---|---|
| 1. Forgetfulness | Strategies to cope with forgetfulness include reminders, habit-based strategies and/or involvement of family members, and improving knowledge on what to do when a dose is forgotten.[23] The use of a reminder tool or pill boxes will be encouraged and a reminder app can be implemented if the patient owns a mobile phone. The habit-based strategy will be delivered through a personalised leaflet, which is tailored to the patient's daily routine (online supplementary appendix C). Patients will be asked to identify the appropriate place and time to take their medication, and an activity they conduct every day that could serve as a prompt or cue to take their medication.[31] Patients will be asked to write coping plans to formulate their own 'if–then' plans for the daily doses of their antihypertensive drug(s).[31] Moreover, patients will be asked to choose a family member to become their treatment supporter and to write down the name of a family member on the personalised leaflet. This individual will be asked to support the patient to take antihypertensive drugs. Pharmacists will keep a copy of the leaflet and remind patients to take his/her personalised leaflet to the next visit. |
| 2. Lack of knowledge | Patient counselling by the pharmacist to cope with lack of knowledge may focus on educating the patient about the purpose of the medication, when and how to take the medication, the need for long-term use, the importance of medication adherence and how to deal with possible side effects. To explore which education is needed, the patient will be asked whether they know why and how to take their medication. The teach-back method will be used, where the patient is asked to explain to the pharmacist what he/she has understood after receiving the education.[32] |
| 3. Lack of motivation | Counselling to cope with a lack of motivation will focus on exploring and discussing the patients' concerns and necessity beliefs. This method is called motivational interviewing.[33] This is done by asking the first question about whether the medication bothers the patient. Follow-up on this question can focus on reducing any concerns or low necessity beliefs (eg, when patients are bothered by the medication because they think the medication is not needed or are afraid of side effects). |
| 4. Other drug-related problems | Counselling to address other drug-related problems will focus on exploring other problems underlying non-adherence, for example, experiencing side effects, costs, polypharmacy, difficulty to refill antihypertensive drugs in time or medication intake problems, and offering solutions/alternatives when possible. |

and recommended intervention strategies are specified in table 1. The session will end with involving patients in goal setting and writing the agreed goal at the top of the personalised leaflet. Pharmacists will remind patients to take his/her leaflet to the next visit.

### Interventions at follow-up (second session)

The follow-up session will be conducted 1 month after the baseline session, when patients refill their medication at the next regular outpatient visit (figure 3). The purpose of the follow-up session is (1) to evaluate the short-term effect of the intervention and discuss the patients' implementation of and experiences with the offered information and recommendations, and (2) to address non-adherence problems that were not yet addressed during the first session. Where needed, the pharmacist, together with the patient, can make changes to the coping plan and discuss additional interventions. The session will end with involving patients in goal setting and writing the agreed goal at the top of the personalised leaflet.

### Pharmacist training

As the quality of the intervention will depend on the competences and skills of the pharmacist, treatment integrity will be enhanced by an obligatory communication training focusing on how to elicit and classify barriers

to adherence, the teach-back method and motivational interviewing, and by providing support material as part of the intervention (figure 4).

Pharmacists and patients participating in this study will be compensated with a modest souvenir at the end of the study for their time and effort.

### Control group

Patients in the five CHCs randomised to the control group will receive pharmacist counselling based on the Indonesian guideline of pharmacy practice (PMK No.74/2016).[19] At each visit, they can receive information about the quantity and dose of the dispensed drugs, when and how to use and store the drugs, side effects and how to deal with them, the importance of medication adherence, and confirming if the patient understands how to take medications correctly. Patients in the control group who were screened as non-adherent to their antihypertensive drugs by the research assistant at baseline will complete the assessments at the same time points as those in the intervention group.

### Outcomes

#### Primary outcome

The primary outcome is the difference between the intervention and the control group in the change in the

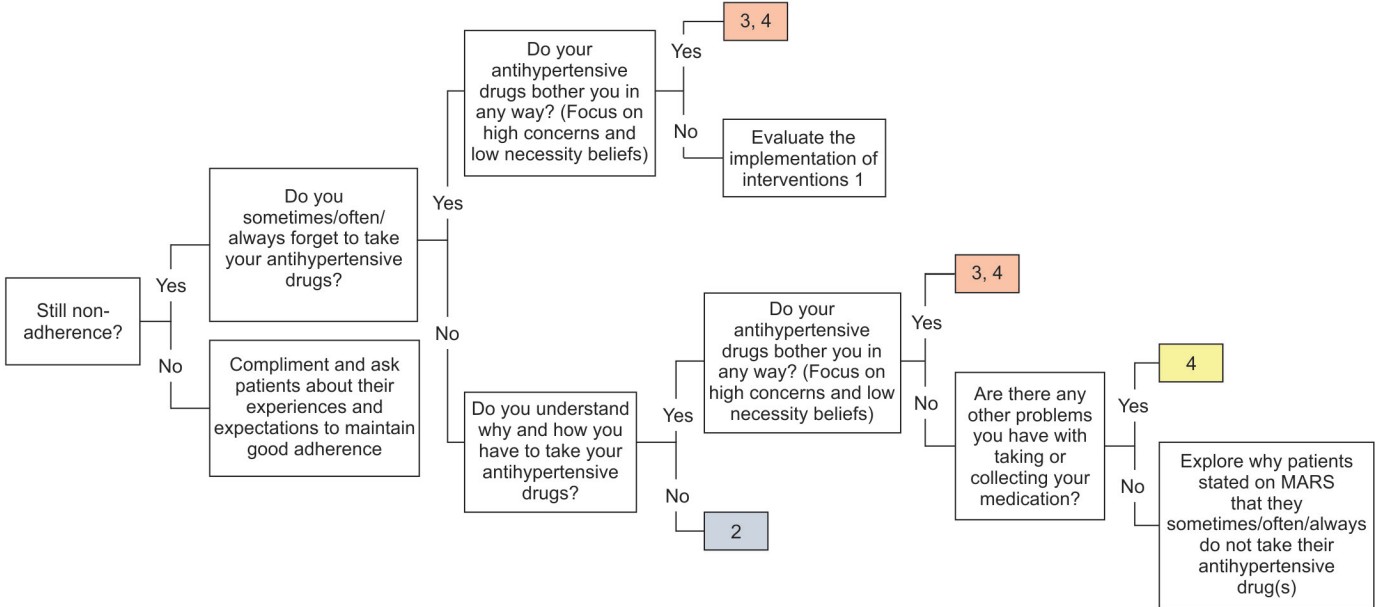

**Figure 3** Flowchart of a targeted and tailored intervention at the 1-month follow-up visit (T1). Type of intervention: 2=counselling to increase knowledge (teach-back method); 3=counselling to increase motivation; 4=explore/address other drug related problems. MARS: Medication Adherence Report Scale.

MARS score from baseline (T0) to a 3-month follow-up (T2). The Indonesian version of the MARS showed to be valid (correlation value of each question to the total score >0.396) and reliable (Cronbach α coefficient of 0.803).[25] Patients will indicate how often each statement applied to them in the last 3 months on a 5-point Likert scale ranging from always (score 1) to never (score 5). Items are summed to obtain a total score ranging from 5 to 25 (Horne R. The Medication Adherence Report Scale (MARS): a new measurement tool for eliciting patients' reports of non-adherence).

### Secondary outcomes
#### Blood pressure level
Within and between patient changes in blood pressure (BP) level (systolic blood pressure and diastolic blood pressure) will be assessed. BP measurements will be performed by a nurse who is blinded to the group assignment at baseline (T0), 1-month (T1) and 3-month (T2) follow-up.

#### Medication beliefs
Within patient, changes on beliefs about the medication will be assessed using the Beliefs about Medicines Questionnaire (BMQ)-specific at baseline (T0) and a 3-month follow-up (T2). The Indonesian version of the BMQ-specific showed to be valid (correlation value of each question to the total score >0.530) and reliable (Cronbach α coefficient of 0.835 and 0.811 for necessity and concerns beliefs, respectively). The BMQ-specific contains five items about necessity beliefs (eg, 'My health at present depends on my blood pressure-lowering medicines'), five items about concern beliefs (eg, 'I sometimes worry about becoming too dependent on my blood pressure-lowering medicines') and one item about side effects (eg, 'My

blood pressure-lowering medicines gives me unpleasant side effects'). All items have a 5-point Likert scale ranging from strongly disagree to strongly agree with an overall range from 5 (low necessity, low concern) to 25 (high necessity, high concern). A necessity–concern differential score will be calculated by subtracting the scores of the concerns scale from the necessity scale (range −20 to 20). A positive differential score indicates stronger beliefs in the necessity, whereas a negative score indicates stronger concerns.[26] The additional item about side effects will be analysed separately due to its known role in non-adherence.[22 27]

#### Process evaluation
We will conduct a process evaluation to assess other parts of the RE-AIM framework. In short, the RE-AIM framework has been developed to evaluate public health interventions assessing five dimensions (reach, efficacy, adoption, implementation and maintenance) at multiple levels (eg, individual or organisation).[18] *Reach* will be assessed by measuring the participation rates and representativeness of patients who participate in this study. In case a patient refuses or discontinues to participate in this study, the patient's age, gender and BP-lowering drugs the patient uses will be recorded by research assistants. This information is used to calculate the participation rate and assess differences between responders and non-responders. To determine representativeness, the patients' demographics will be compared with census demographics in Bandung City, Indonesia. *Adoption* will be evaluated by assessing the proportion and representativeness of CHCs who participate in this study, and exploring pharmacists' and patients' satisfaction with and willingness to use various parts of the intervention.

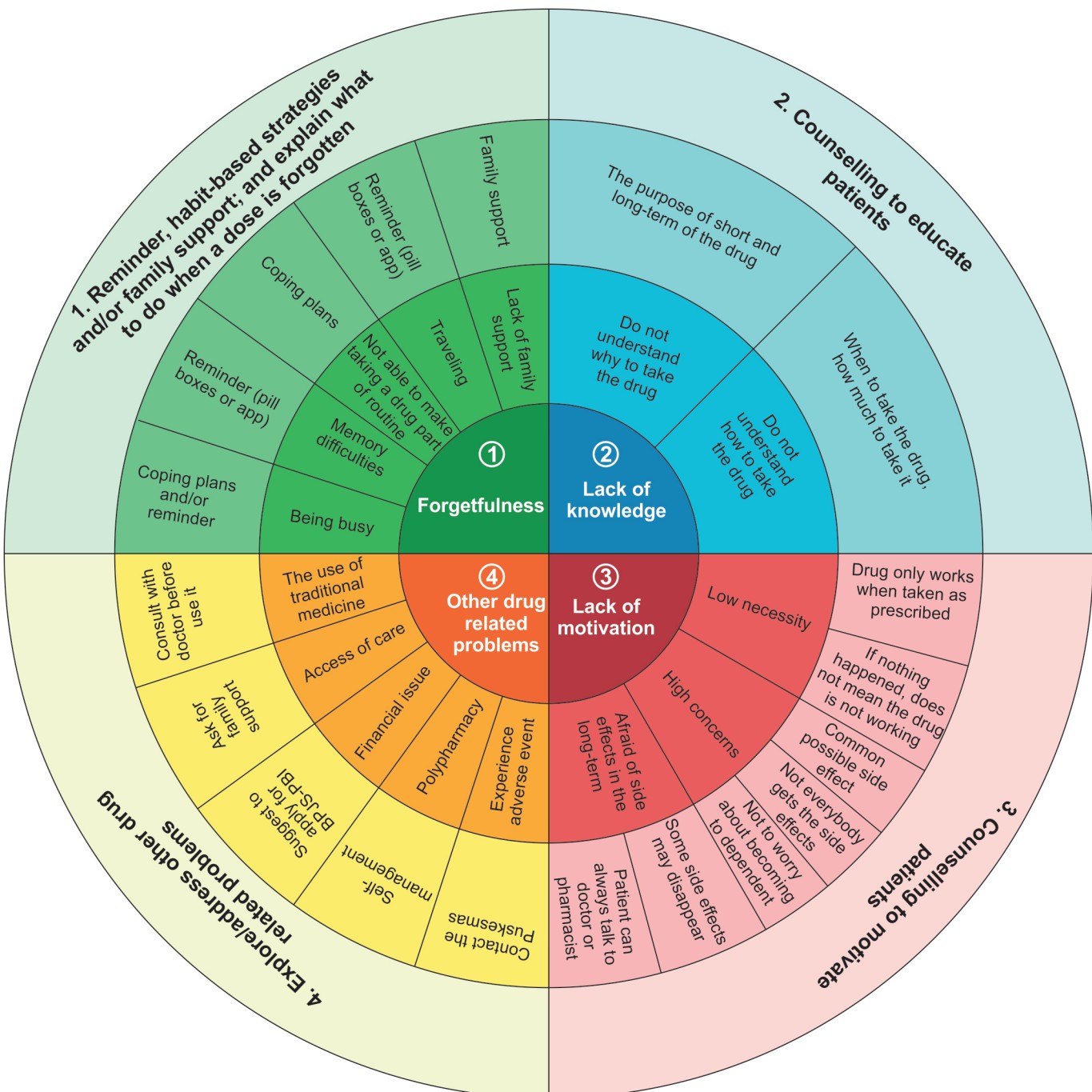

**Figure 4** Proposed adherence intervention wheel as supportive material for pharmacists in the intervention group.

*Implementation* will be evaluated by determining whether the intervention was delivered as intended and exploring pharmacists' and patients' suggestions for future implementation. *Maintenance* will be assessed by determining whether the intervention can be maintained and the willingness of pharmacists and payers to continue the intervention as part of routine clinical practice. We will use focus group discussions at a 1-month (T1) follow-up and an evaluation survey (based on a previously used survey)[28] at a 3-month follow-up (T2) to explore pharmacists' and patients' adoption, implementation and willingness to maintain the intervention, respectively.

**Baseline participant and CHC characteristics**

Participants' baseline characteristics, including sociodemographic and clinical-related factors, will be obtained. Sociodemographic factors are self-reported and include age at the completion of the questionnaire, gender, highest level of education completed (no formal education/elementary high school, junior high school, senior high school or university) and type of health insurance. Type of health insurance will be classified as those whose insurance premium was paid by the government (BPJS-PBI), those whose insurance premium was paid by the patients themselves (BPJS-Non PBI) or those without

health insurance. Clinical-related factors include time since diagnosis of diabetes and hypertension (years), diabetes complication(s) that developed after the diagnosis of diabetes, and types and number of concomitant medications. The following diabetes complications will be considered: cardiovascular conditions, cerebrovascular conditions, nephropathy, retinopathy, neuropathy and diabetic foot problems. Clinical-related factors will be collected by research assistants using a predefined data collection form. Furthermore, organisational information of each CHC (number of medical doctors, pharmacists, nurses and average number of patients with diabetes with and without hypertension visits per month) and pharmacist characteristics (age, gender and working experience in community pharmacy (years)) will be collected by research assistants using a predefined data collection form.

### Treatment fidelity

Treatment fidelity will be addressed by providing a checklist of items that pharmacists need to do at each patient visit and a counselling protocol for the intervention group (online supplementary appendix D). Pharmacists will be asked to complete the checklist after each visit with a study participant. The completed checklists will be collected on a weekly basis and used to calculate an overall fidelity score. Minor feedback suggestions from the PI to pharmacists will be made if needed.

### Sample size calculation

The sample size calculation is based on the formula for cluster randomised trials, powered on the primary outcome.[29] We want to be able to detect a difference between the intervention and control group in the change in adherence score of at least 2.5 points with an expected SD of 3.8 points and assuming an intracluster correlation coefficient within CHCs of 0.014, as calculated based on the previous work from our group (manuscript in preparation). A sample size of 41 non-adherent patients in each study arm (intervention and control group) would allow for 80% power to detect this difference using a two-sided test at the 5% level of significance. Assuming non-adherence rates of 50%[30] and a dropout rate of 20%, we will recruit at least 100 patients in each group, giving a total of 200 patients from 10 CHCs (20 per CHC) that need to complete the MARS screening. Recruiting at least 20 patients per CHC is feasible as the average number of diabetes patients with hypertension visiting a CHC is around 30 patients per month.

### Planned statistical analysis

Data analysis will be performed based on the intention-to-treat principle. Descriptive statistics will be used to summarise the baseline characteristics. To control for the effects of cluster randomisation, group changes will be compared (individual CHCs will be treated as a random effect) using multivariate mixed linear and non-linear regression for normally and non-normally distributed data, respectively. Point estimates estimated from cluster-adjusted models will be reported with 95% confidence intervals . We will conduct subgroup analysis using data stratification on diabetes complications and the number of concomitant medications to assess the effect of the intervention. All tests will be two-tailed and p<0.05 will be considered statistically significant. All statistical analyses will be carried out using SPSS software (V.25).

### Data management

Data from the questionnaires and case report forms will be entered by the PI using a unique identifier that is provided for each participant into SPSS software V.25 and data forms will be stored on a password-protected computer.

### Adverse event reporting

It is possible that a participant identifies a medication-related issue during pharmacist counselling, either in the intervention or control group. Although this is unlikely to be a result of the study, the patient's doctor will be contacted if the pharmacist and/or researchers have concerns requiring immediate intervention.

### Patient and public involvement

Patients and the public were not involved in the development of the research question or outcome measures. Patients will be involved during the conduct of the study by giving feedback to tailor the intervention based on their personal adherence barrier(s). In addition, patients will be asked to complete an evaluation survey, including questions about the intervention. Patients will be given contact details of the PI to request the results of the study.

## ETHICS AND DISSEMINATION

All results will be stored securely and will be available to authorised individuals for analysis and reporting purposes only. Data will be published in a form that does not identify patients in any way. To maintain patients' anonymity, a unique identifier will be used to match patients' data across baseline and follow-up. Patients are free to withdraw from the study at any time. The findings of the study will be disseminated through international conferences, one or more peer-reviewed journals, and reports to key stakeholders.

**Author affiliations**
[1]Unit of Pharmaco-Therapy, -Epidemiology & -Economics, Department of Pharmacy, Groningen Research Institute of Pharmacy, University of Groningen, Groningen, The Netherlands
[2]Department of Pharmacology and Clinical Pharmacy, Faculty of Pharmacy, Universitas Padjadjaran, Jatinangor, Indonesia
[3]Center of Excellence in Higher Education for Pharmaceutical Care Innovation, Universitas Padjadjaran, Jatinangor, Indonesia
[4]Department of Clinical Pharmacy and Pharmacology, University Medical Centre Groningen, University of Groningen, Groningen, The Netherlands
[5]Medication Adherence Expertise Center Of the northern Netherlands (MAECON), Groningen, The Netherlands

**Contributors** SDA: wrote the first draft of this protocol. SDA, RA, PD, JFMvB, and EH: participated in the design of the study and contributed to the revision of the study protocol. All authors approved the final manuscript.

**Funding** SDA is supported by a scholarship from the Indonesia Endowment Fund for Education (LPDP No: PRJ-2361/LPDP/2015).

**Disclaimer** The Indonesia Endowment Fund for Education has no role in designing the study, in writing this article, and in deciding to submit it for publication.

**Competing interests** None declared.

**Patient consent for publication** Obtained.

**Ethics approval** Ethical approval for this study was obtained from the Ethical Committee of Universitas Padjadjaran, Indonesia (No. 859/UN6.KEP/EC/2019).

**Provenance and peer review** Not commissioned; externally peer reviewed.

**ORCID iD**
Sofa D Alfian http://orcid.org/0000-0001-5419-8938

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
