## [Reviewer comments · BMJ Open]

ARTICLE DETAILS

TITLE (PROVISIONAL)	Targeted and tailored pharmacist-led intervention to improve adherence to antihypertensive drugs among patients with type 2 diabetes in Indonesia: study protocol of a cluster randomized controlled trial
AUTHORS	Alfian, Sofa; Abdulah, Rizky; Denig, Petra; van Boven, Job; Hak, Eelko

VERSION 1 – REVIEW

REVIEWER	Lise Aagaard University of Copenhagen, Department of Pharmacology and Pharmacotherap
REVIEW RETURNED	28-Oct-2019

GENERAL COMMENTS	Thank you for the opportunity to review this protocol. The authors should consider extending the follow up-period from 3 to 6 months as adherence might change over time, and particularly after 3 months.
--

REVIEWER	Stefano Omboni Italian Institute of Telemedicine, Italy
REVIEW RETURNED	31-Oct-2019

GENERAL COMMENTS	In this protocol paper, the authors describe the design of a randomized controlled study, aiming at evaluating the impact on low adherence of a low-cost, targeted and tailored pharmacist-led intervention integrated into a community pharmacy workflow in Indonesia. In this study type 2 diabetics under antihypertensive treatment will be recruited. The study protocol refers to an ongoing study, commenced in August 2019 and ending in December 2019. A sample size estimation has been done, based on a previous work of the authors, whose manuscript is in preparation. However, there are several published studies and meta-analyses showing the impact of the pharmacist's intervention on adherence in diabetics/hypertensives and thus the authors may have used these studies to try to elaborate a more meaningful hypothesis based on more realistic numbers. I recommend to search for the published literature and make a proper sample size estimation. The inclusion criteria do not foresee a selection based on low adherence (page 6, lines 52-59), though authors specify on page 7 that only low-adherent subjects (MARS score <20) will be included. Thus, the list of inclusion criteria must be updated and better detailed.
---

	Inclusion criteria. A generic diagnosis of type 2 diabetes may not suffice. These patients may have been untreated, or following a diet, or taking blood glucose-lowering drugs or insulin. Differences in the severity of diabetes and intensity of treatment can influence the adherence to antihypertensive treatment. The authors should better detail inclusion/exclusion criteria. If patients with several comorbidities are accepted (see page 13 when describing diabetes complications) proper adjustment or stratification should be foreseen, because adherence may vary according to number of co-morbidities and complexity of drug treatment. Inclusion/exclusion criteria. Apparently, the authors do not specify if the presence of comorbidities such as, for instance, cardiovascular or kidney disease, is considered (see again reference to diabetes complications on page 13). Again, this has an impact on adherence, due to number of pills, other antihypertensive medications. Inclusion criteria. The generic selection of treated hypertensive patients may be dangerous because the adherence depends on the number of drugs and whether a patient is resistant. The authors should have focused on specific BP thresholds and number of drugs otherwise the population is too heterogeneous and the numbers foreseen may not be sufficient. Why the authors did not consider to assess also adherence to antidiabetic therapy (if any) and investigate any influence on adherence to antihypertensive drug treatment? The study does not foresee any contact between patients and pharmacists between visit at 1-month and the final visit after 3-months, for both study groups. Given the short-lasting of the study, patients may have benefit of an additional visit at month 2 in order to better evaluate the MARS score.
--	---

VERSION 1 – AUTHOR RESPONSE

Reviewer(s)' Comments to Author:

Reviewer 1

Thank you for the opportunity to review this protocol. The authors should consider extending the follow up period from 3 to 6 months as adherence might change over time, and particularly after 3 months.

RESPONSE: Thank you for your suggestion. Indeed, we agree that adherence might change over time. Yet, we first would like to assess the immediate effects of the intervention. If this intervention is found to be effective, follow-up research may be conducted, taking into account that adherence might change over time.

Reviewer 2

In this protocol paper, the authors describe the design of a randomized controlled study, aiming at evaluating the impact on low adherence of a low-cost, targeted and tailored pharmacist-led intervention integrated into a community pharmacy workflow in Indonesia. In this study type 2

diabetics under antihypertensive treatment will be recruited. The study protocol refers to an ongoing study, commenced in August 2019 and ending in December 2019.

1. A sample size estimation has been done, based on a previous work of the authors, whose manuscript is in preparation. However, there are several published studies and meta-analyses showing the impact of the pharmacist's intervention on adherence in diabetics/hypertensives and thus the authors may have used these studies to try to elaborate a more meaningful hypothesis based on more realistic numbers. I recommend to search for the published literature and make a proper sample size estimation.

RESPONSE: Thank you for your suggestion. We are aware of the existing meta-analyses. However, most of the underlying studies in these meta-analyses have been performed in high-resource settings that are not representative for our setting. For example, a recent meta-analysis conducted by Conn e.a. 2017 included only 12% of studies conducted in Asia, many of which focused on HIV/AIDS/tuberculosis in India or China. This meta-analysis showed huge differences in effect sizes between regions, illustrating the relevance of using local data. In addition, this and other meta-analyses have shown that the expected effect size will depend on the adherence measurement used as well as the disease focus. Therefore, we prefer to use data that are more representative for our setting and study population to estimate the adherence and standard deviation as well as the intra-cluster correlation coefficient (ICC) for the sample size calculation, based on our previous study in community health centers (CHCs) in Indonesia using the same measurement instrument.

2. The inclusion criteria do not foresee a selection based on low adherence (page 6, lines 52-59), though authors specify on page 7 that only low-adherent subjects (MARS score <20) will be included. Thus, the list of inclusion criteria must be updated and better detailed.

RESPONSE: Thank you for this remark. We have updated our inclusion criteria including a selection based on low adherence score (page 7).

3. Inclusion criteria. A generic diagnosis of type 2 diabetes may not suffice. These patients may have been untreated, or following a diet, or taking blood glucose-lowering drugs or insulin. Differences in the severity of diabetes and intensity of treatment can influence the adherence to antihypertensive treatment. The authors should better detail inclusion/exclusion criteria. If patients with several co-morbidities are accepted (see page 13 when describing diabetes complications) proper adjustment or stratification should be foreseen, because adherence may vary according to number of co-morbidities and complexity of drug treatment.

RESPONSE: Thank you for your comment. Although our main goal is to assess the effects of the intervention in a population-based, heterogeneous, type 2 diabetes population, the intervention may indeed have different effects in subgroups within this study population. Following your comment, we now include additional subgroup analyses using data stratification on diabetes complications and number of concomitant medications. We have updated our objectives (page 5) and adjusted our planned statistical analysis (page 15).

4. Inclusion/exclusion criteria. Apparently, the authors do not specify if the presence of comorbidities such as, for instance, cardiovascular or kidney disease, is considered (see again reference to diabetes complications on page 13). Again, this has an impact on adherence, due to number of pills, other antihypertensive medications.

RESPONSE: As mentioned in the response to point 3, we now include subgroup analysis and have updated our objectives (page 5) and adjusted our planned statistical analysis (page 15) accordingly.

5. Inclusion criteria. The generic selection of treated hypertensive patients may be dangerous because the adherence depends on the number of drugs and whether a patient is resistant. The authors should have focused on specific BP thresholds and number of drugs otherwise the population is too heterogeneous and the numbers foreseen may not be sufficient.

RESPONSE: We thank the reviewer for mentioning the impact of number of drugs used. Therefore, data stratification based on number of drugs will be performed and this has been specified in the revised manuscript (see also response to point 3 and 4). Regarding the focus on specific BP thresholds for inclusion criteria instead of non-adherence level, this has been considered. Yet, in the light of the core components of our intervention program, it was deemed inefficient and costly to include patients who have no or only minor problems in being adherent with their treatment. We anticipate that our pharmacist-led intervention should work in a heterogeneous population regardless of exact BP threshold, though BP level will be included as a secondary outcome (see page 11). It uses principles of targeting by screening for non-adherence and tailoring on patient-specific information. Regarding the sample size, this was estimated based on data from a similar heterogeneous population (page 14).

6. Why the authors did not consider to assess also adherence to antidiabetic therapy (if any) and investigate any influence on adherence to antihypertensive drug treatment?

RESPONSE: Our primary question is to assess the effect of a tailored and targeted intervention on adherence to antihypertensive drug treatment. Adherence to antihypertensive drug treatment is a huge problem in the Indonesian setting. Our previous study showed that around half of patients with diabetes in Indonesia were non-adherent to antihypertensive drugs (45.5%, 492 patients). Adherence to other drugs, including antidiabetic therapy, may be associated with adherence to antihypertensive drug treatment but in a RCT we expect that the influence of other factors is similar in both study arms. Furthermore, assessing adherence to other drugs would create more burden for the patients with the risk of more non-response and loss to follow-up.

7. The study does not foresee any contact between patients and pharmacists between visit at 1-month and the final visit after 3-months, for both study groups. Given the short-lasting of the study, patients may have benefit of an additional visit at month 2 in order to better evaluate the MARS score.

RESPONSE: Indeed, we agree that adding an additional visit at month 2 might provide more benefit but may not necessarily be more cost-efficient. Adding another visit would create an approach that is not considered feasible to implement in daily practice in Indonesia. Since we aimed to develop a low-cost and pragmatic pharmacist-led intervention that can be integrated into the community pharmacy workflow in Indonesia, our intervention is limited this to the 1-month visit.

Reference

Conn VS, Ruppert TM. Medication adherence outcomes of 771 intervention trials: systematic review and meta-analysis. Preventive medicine. 2017 Jun 1; 99:269-76.

VERSION 2 – REVIEW

REVIEWER	Stefano Omboni Italian Institute of Telemedicine, Clinical Research Unit
REVIEW RETURNED	28-Nov-2019
GENERAL COMMENTS	I am satisfied with the changes implemented in the paper by the authors. However, I see that on page 5 they have removed the distinction between primary and secondary objectives. I

	recommend leaving this distinction because only the sample size estimation is referred to the primary endpoint and the secondary endpoints may not be powered enough.
--	---